# Effect of Propolis on Root Dentine Microhardness When Used as an Intracanal Medicament: An In Vitro Study

**DOI:** 10.3390/jfb14030144

**Published:** 2023-03-03

**Authors:** Meshal Muhammad Naeem, Huma Sarwar, Aliza Nisar, Shahbaz Ahmed, Juzer Shabbir, Zohaib Khurshid, Paulo J. Palma

**Affiliations:** 1Department of Science of Dental Materials, Dr Ishrat Ul Ibad Khan Institute of Oral Health Sciences, Dow University of Health Sciences, Karachi 74200, Pakistan; 2Department of Operative Dentistry, Dr Ishrat Ul Ibad Khan Institute of Oral Health Sciences, Dow University of Health Sciences, Karachi 74200, Pakistan; 3Karachi Medical and Dental College, Karachi 75340, Pakistan; 4Department of Operative Dentistry & Endodontics, Baqai Dental College, Karachi 75340, Pakistan; 5Department of Prosthodontics and Implantology, College of Dentistry, King Faisal University, Al Hofuf 31982, Saudi Arabia; 6Center of Excellence for Regenerative Dentistry, Department of Anatomy, Faculty of Dentistry, Chulalongkorn University, Bangkok 10330, Thailand; 7Center for Innovation and Research in Oral Sciences (CIROS), Faculty of Medicine, University of Coimbra, 3000-075 Coimbra, Portugal; 8Institute of Endodontics, Faculty of Medicine, University of Coimbra, 3000-075 Coimbra, Portugal

**Keywords:** calcium hydroxide, dentine microhardness, intracanal medicament, propolis

## Abstract

Application of intracanal medicaments may affect the physical properties of root dentine. Calcium hydroxide (CH), a gold standard intracanal medicament, has proven to decrease root dentine microhardness. A natural extract, propolis, has been shown to be superior to CH in eradicating endodontic microbes, but its effect on the microhardness of root dentine is still not known. This investigation aims to evaluate the effect of propolis on root dentine microhardness compared to calcium hydroxide. Ninety root discs were randomly divided into three groups and treated with CH, propolis, and a control. A Vickers hardness indentation machine with a load of 200 g and dwell time of 15 s at 24 h, 3, and 7 days was used for microhardness testing. ANOVA and Tukey’s post hoc test were used for statistical analysis. A progressive decrease in microhardness values was observed in CH (*p* < 0.01), whereas a progressive increase was observed in the propolis group (*p* < 0.01). At 7 days, propolis demonstrated the highest microhardness value (64.43 ± 1.69), whereas CH demonstrated the lowest value (48.46 ± 1.60). The root dentine microhardness increased over time when propolis was applied, while it decreased over time after application of CH on root dentine sections.

## 1. Introduction

The removal of bacteria and their toxic byproducts from the root canal system is essential for successful endodontic therapy [1]. In order to render the infected root canals sterile, they are mechanically and chemically debrided using different instruments and reagents. The most essential step in successful endodontic therapy is the complete debridement of the whole endodontic space and permanent sealing of the root canal space. The removal of endodontic microbes and their toxic byproducts from the endodontic space is essential for good prognosis of the endodontic treatment [2]. Shaping of the root canals either manually or with rotary endodontic instruments not only removes infected and necrosed pulp but also prepares the root canal for facilitation of irrigation, medicaments’ placement, and obturation. Endodontic cleaning is performed by irrigation solutions, being effective against endodontic microbes, removing dentinal debris, and lubricating the root canals. Cleaning and shaping procedures ensure bulk bacteria and their toxins’ removal. Intracanal medicaments are primarily used during multi-visit endodontic treatment to reduce microbial proliferation between appointments. Intracanal medicaments play avital role in rendering the root canal system sterile when merely chemo-mechanical debridement is insufficient to counter surviving resistant bacteria that can proliferate between appointments [3]. These medicaments are not only effective against surviving microorganisms but also minimize the ingress of pathogens from leaky restorations [4]. The placement of intracanal medicaments inhibits proliferation of surviving bacteria, supplies continued disinfection, and creates a physical barrier against bacterial reinvasion between appointments [5]. Intracanal medicaments are the endodontic materials with good biocompatibility, placed temporarily to control proliferation microbes within the root canal system and to prevent invasion of bacteria from the oral cavity. Ideally, an intracanal medicament should have desired antimicrobial and anti-inflammatory effects, provide a physical barrier for bacterial invasion, and can be easily placed and removed. It should be non-toxic and biocompatible. It should not adversely affect the physical and mechanical properties of root dentine, such as microhardness, modulus of elasticity, and flexural strength, and must not be cytotoxic to periapical tissues. On the contrary, some of the intracanal medicaments may negatively affect important physical properties of root dentine, such as microhardness. Microhardness is a measurement indicating mineral gain or loss in dental hard tissues and is measured by an indenter that penetrates microscopic areas [6]. Root dentine microhardness is often correlated with mineral concentration. Decreased microhardness reflects demineralization of root dentine, and vice versa. A decrease in root dentine microhardness reflects softer root dentine that negatively affects the sealing ability of obturation materials, leading to compromised prognosis of the endodontically treated tooth [7,8].

Calcium hydroxide (CH) has been used in dentistry, especially endodontics, since the 1920s. CH is a crystal, poorly soluble in water, and induces only localized effects. CH can be mixed with saline or sterile water and is also commercially available in sterile single-dose packages. The slurry of calcium hydroxide is best applied with lentulospiral after root canal preparation. CH is normally used as a slurry in a water base. Less than 0.2% of CH is dissolved into Ca^2+^ and OH^−^ ions at body temperature. Water should be the vehicle as CH needs water to dissolve. CH is a slow-acting antiseptic. In addition to its antibacterial property, CH also hydrolyzes the lipid moiety of bacterial lipopolysaccharide, and therefore inactivates the biological activity of the LPS and reduces its effect [9]. CH is known to exhibit antibacterial, anti-inflammatory, and tissue-dissolving properties and has shown osteogenic potential as well [10]. Unlike some of the conventionally used materials such as phenolics and aldehydes, CH does not induce strong local or systemic harmful effects [7,11]. Due to its high pH, CH is capable of eliminating specific bacteria from the root canal system when used for up to seven days [6] but has been found to be ineffective against *E*. fecalis, the most commonly inoculated bacteria in failed endodontic canal treatments [12]. Furthermore, the literature also reports the negative impact of CH on root dentine microhardness when used as an intracanal medicament [7,11]. These potential side effects of CH have led to the recent popularity of natural alternative endodontic medicaments. These include propolis, Aloe Vera (*Aloe barbadensis miller*), Curcuma longa (*Turmeric*), Salvadora Persica Solution (*Miswak-siwak*), Acacia nilotica (*Babool*), Morinda Citrifolia (*noni*), chitosan, garlic (*Allium satium*), and Triphala and green tea polyphenols. Natural products are known for their high antibacterial, biocompatible, antioxidant, and anti-inflammatory properties [13].

Propolis is a complex resinous mixture which contains approximately 50% resin and balsam, 30% wax, 10% essential and aromatic oils, 5% pollen, and 5% impurities [14]. Flavonoids in propolis are responsible for its antimicrobial properties. Several studies have demonstrated the antibacterial properties of propolis and recommended its use as an intracanal medicament [15,16]. Besides the biological role it plays, the effect of propolis on root dentine microhardness is not known. Propolis has proven to increase enamel microhardness when applied to the external tooth surface [17]. It has also proven to increase root dentine microhardness when used as an endodontic irrigant [18]. However, to date, the effect of propolis, as an intracanal medicament, on root dentine microhardness has not been reported and no statistical comparison of the effect of propolis and CH on root dentine microhardness has been evaluated. Therefore, this study aimed to compare the microhardness of root dentine when it is exposed to CH and propolis for up to seven days. According to the null hypothesis, there will be no difference in root dentine microhardness after application of intracanal medicaments at different time intervals.

## 2. Materials and Methods

This in vitro randomized controlled trial was approved by the Institutional Review Board of Dow University of Health Sciences (IRB-797/DUHS/Approval/2016/326). In this study, root dentine discs were prepared from extracted teeth. Mature, permanent, single-rooted teeth from either arch (mandibular or maxillary) of 18- to 40-year-old individuals were included in this study. Teeth with resorptions, decay, or fracture below the cementoenamel junction, developmental defects, or with previous root canal treatment were excluded from the research. The non-probability (purposive) sampling technique was used for sample recruitment and the simple random method was used for group allocation using MS Excel software. Sample size calculation was carried out using PASS software at 80% power of the test, a 95% confidence interval, and a 5% level of significance. The sample size was calculated by taking the mean and standard deviations of three groups (52.03 ± 1.23, 54.67 ± 1.63, and 55.06 ± 3.36) [6], and the initially calculated sample size was 11 root discs in each group. To increase the statistical significance, the sample size was increased from 11 to 30 root discs in each group. Forty-five extracted teeth were used to make ninety root discs.

### 2.1. Preparation of Medicaments

#### 2.1.1. Calcium Hydroxide (CH)

To make CH paste, on a clean sterile glass slab, a 1.5:1 ratio (wt/vol) of CH powder was mixed with sterile saline until the desired consistency of CH paste was attained [6].

#### 2.1.2. Propolis

For the preparation of propolis paste, on a clean sterile glass slab, 95% propolis powder (Henan Fumei Bio-Technology Co., Ltd., Zhengzhou, China, Reg No. 411082100010933) was mixed with saline to make a paste of the desired consistency.

### 2.2. Preparation of Root Dentine Discs

Forty-five extracted teeth were stored in 0.1% thymol at room temperature before use. Soft tissue remnants, debris, and calculus were removed by an ultrasonic scaler. Crowns of the teeth were removed using a diamond disc and pulp was extirpated with a #10 k-file (Dentsply Tulsa Dental Specialties, Johson City, TN, USA). Root canals were prepared using the Protaper Universal filing system (Dentsply Tulsa Dental Specialties, Johson City, TN, USA) until File F3. Irrigation was performed during instrumentation with 3% sodium hypochlorite (CanasolTM, Islamabad, Pakistan). Roots were then embedded in acrylic resin and cut transversally into coronal third, middle third, and apical third sections using a band saw. Coronal and apical thirds of the roots were discarded and a root section of 4 mm from the middle third of each root was used to obtain two root dentine discs of 2 mm each (Figure 1). A Vernier caliper was used for the measurement of root disc dimensions. Specimens were polished with 2500-grit abrasive paper (Hermes, Hamburg, Germany) using a micro-grinding system (Exakt 400 cs pparatebau, Norderstedt, Germany).

### 2.3. Data Collection Procedure

Ninety root discs were prepared and randomly divided into three groups: Group 1: calcium hydroxide (CH), Group 2: propolis, and Group 3: control group. Root disc specimens were prepared and placed in the intracanal medicaments (according to the assigned groups) in airtight containers at 37 °C and 100% humidity in an incubator and subjected to microhardness testing after 1, 3, and 7 days.

### 2.4. Microhardness Assessment

After treatment with medicaments, the specimens were rinsed with distilled water and dried with absorbent paper before measuring microhardness. Microhardness testing was performed after 1, 3, and 7 days with a Vickers hardness indentation machine (Future-Tech Corp FM-700, Tokyo, Japan). The indentations were made with a Vickers hardness intender at 40× magnification and recorded as 3 separate indentations at a depth of 100 µm each using a 200 g load and a 15 s dwell time, with a distance of at least 500 μm between indentations. The indentations were placed at 1 mm from the root canal wall. The length of the two diagonals was used to calculate the microhardness value (Vickers hardness number (VHN)). Root dentine microhardness was measured at three different points in each root disc and the mean was calculated. A control group was used to record baseline data. Values were recorded as Vickers hardness number (VHN). Data were recorded, tabulated, and statistically analyzed. Microhardness testing, measurements, and recording on the data collection form were performed by a blinded examiner.

### 2.5. Statistical Analysis

Data were entered and analyzed using SPSS version 21.0. The significance level was set at α = 0.05. Analysis of variance (ANOVA) was used for intragroup comparison to identify significant mean difference of Vickers hardness number (VHN) among all three groups. Furthermore, Tukey’s post hoc test was used for intergroup comparisons of mean VHN differences at different time intervals. For the convenience of the readers, a summary of the whole methodology is provided in Figure 2.

## 3. Results

Table 1 demonstrates the intragroup comparison of microhardness values at three different time intervals. CH and propolis showed a statistically significant difference between 1, 3, and 7 days (*p* < 0.01). CH demonstrated a progressive decrease in microhardness values over time (*p* < 0.01), whereas propolis showed a progressive increase in microhardness values over time (*p* < 0.01). No significant difference in microhardness values was observed at different time intervals in the control group (*p* < 0.26).

The intergroup comparison of microhardness mean values at different time intervals is demonstrated in Figure 3. At all three time intervals, propolis demonstrated the highest VHN values, followed by control and CH groups, respectively. The highest VHN value (64.43) was demonstrated by the propolis group at 7 days, whereas the CH group at 7 days showed the lowest VHN value (48.46). Furthermore, it can be observed that when CH and propolis were compared, the mean differences in microhardness values progressively increased at 1, 3, and 7 days (−3.16, −9.42, and −15.97, respectively).

Pairwise comparison was carried on using the post hoc test, as shown in Table 2 at 1 day. Table 2 shows that the mean difference of microhardness values (VHN) at 1 day between CH and propolis groups was statistically significant, as the *p*-values were less than 0.05. Intergroup pairwise comparison between CH vs. control and propolis vs. control showed an insignificant difference (*p* = 0.017 and *p* = 0.966, respectively). Comparison between CH vs. propolis showed the highest mean difference in microhardness reduction, followed by CH vs. control and propolis vs. control, respectively.

In Table 3, a pairwise comparison (post hoc test) was carried out, showing that the mean differences in microhardness value reduction for pairs of CH, propolis, and control groups at 3 days of intervention were statistically significant (*p* < 0.01). The results showed that the highest mean difference in microhardness reduction was observed when the groups CH vs. propolis were compared, followed by the comparison between CH vs. control and propolis vs. control, respectively.

In Table 4, the intergroup comparison of mean differences in microhardness values at 7 days of intervention showed statistically significant values for all the group pairs (*p* < 0.05). The comparison between the CH vs. propolis groups showed the highest mean difference in VHN values, whereas propolis vs. control demonstrated the lowest mean difference in microhardness values of root dentine.

The highest mean difference for Vickers hardness was −15.97, observed for CH vs. propolis at 7 days, and the minimum was 0.53 for control at 1 day vs. control at 3 days.

## 4. Discussion

In this ex vivo study, the effect of CH and propolis, when used as an intracanal medicament, was investigated on dentine microhardness after 24 h, 3 days, and 7 days. Based on the contemporary best-available evidence, the gold standard intracanal medicament, CH, has demonstrated limited effectiveness in eliminating microbes from the root canals and negatively affects the root dentine microhardness. As the quest for an ideal intracanal medicament that ensures better disinfection of the root canal space and maintains the mechanical strength of the root dentine continues, this topic was selected to be tested. Root dentine microhardness testing provides an insight into mineral loss or gain as it depends on the amount of calcified matrix per millimeter. Therefore, a decreased root dentine microhardness value demonstrates the softening effect of the intracanal medicament on the root dentine section, and vice versa. As a consequence of decreased root dentine microhardness, this relative softening effect adversely affects the sealing ability of endodontic sealers to the root dentine and eventually compromises the quality of endodontic therapy [19]. To the best of the authors’ knowledge, no study to date has evaluated the effect of propolis when used as an intracanal medicament on the root dentine microhardness in comparison to CH, and therefore this parameter was selected to be tested in this study.

For the measurement of root dentine microhardness, two testing methods are usually employed: Knoop microhardness testing and Vickers hardness testing. In the present study, Vickers hardness testing was employed instead of Knoop testing as the former method is more sensitive to measurement errors, less sensitive to surface conditions, and small specimens such as root discs can be tested with good accuracy [20]. It is a test performed to measure the microhardness of substances and materials, specifically thin sections and small parts. During this test, a light load is applied via a diamond indenter to produce an indentation on the subject under testing. The depth of the indentation is converted into the hardness value of the object. An inverse correlation between tubule density and dentine microhardness has been reported in the past by Pashley et al. [21]. According to their study, as the tubular density increases and the amount of inter-tubular dentine decreases near the pulp chamber, the root dentine microhardness decreases. Microhardness of dentine decreases as the indentations are made closer to the pulp [22]. In the present study, to measure the Vickers hardness values for dentine, indentations were made 1 mm from the root canal walls at three different points and were performed at a depth of 100 µm for standardization, each using a 200 g load and a 15 s dwell time. Lighter load and less of a dwell time were used because of the inverse correlation between dentine microhardness and tubular density [23]. Therefore, all the root dentine blocks were prepared from the middle third of the roots and were standardized to have 2 mm of thickness. Russell et al. [24] reported that teeth with a butterfly effect showed higher dentine tubule density buccolingually than mesiodistally, and the root sections of such teeth demonstrated higher microhardness scores on their mesial and distal surfaces. Therefore, on each root dentine specimen, microhardness was tested at three different points to draw a mean microhardness value. It has also been reported that inter-tubular dentine of teeth belonging to elderly individuals demonstrate increased microhardness [25]; therefore, to control for another confounding factor, the age group was standardized for the patients whose extracted teeth were included in this study to minimize this effect of age on the microhardness scores, thereby excluding the chances of dentine or cementum deposition due to the aging process. The extracted permanent teeth included in this study used for dentine blocks’ preparations were selected from patients aged between 18 and 40 years.

According to the results, the null hypothesis was rejected because the root dentine microhardness decreased over time in the CH group, whereas it increased over time in the propolis group. The control group demonstrated the least effect on microhardness values compared to the other two groups.

The findings of the present study corroborate the results of another study by Yassen [26]. According to that study, a one-week application of CH on dentine resulted in significant demineralization and collagen degradation of radicular dentine.

Furthermore, it has also been proven that the increased pH of CH reduces the organic support of dentine matrix, resulting in collagen fiber link disruption and breakdown of the protein structure, eventually negatively affecting the mechanical properties of the root dentine [27]. This could be a possible explanation for the reduced VHN values in the CH group reported in the present study. Similar results of reduced root dentine microhardness after CH application have been reported by Elfaramawy et al. [28]. According to them, CH reduces dentine microhardness over time, which can be countered by the addition of activated charcoal. Amonkar et al. [29] also reported that long-term application of CH significantly reduces dentine microhardness. A study by Parashar et al. [30] also reported a significant reduction in microhardness values after CH. The findings of all these studies corroborate the results of the present study. It has also been proven that the high alkalinity and small size of CH facilitates its penetration into the intra-fibrillar structure of collagen and leads to alteration of the 3D conformation of tropocollagen, resulting in reduced microhardness and elastic modulus [31]. Consequently, because of the alkaline properties of CH, it has a strong denaturing effect on the organic matrix of dentine [32]. Prabhakar et al. [6] compared another natural extract (turmeric) with CH paste and reported that CH significantly reduced dentine microhardness when compared with turmeric. Similar results have been observed in the present study when CH was compared with propolis.

The literature lacks evidence in reference to propolis having any effect on microhardness of root dentine when used as an intracanal medicament. According to our knowledge, this is the first research reporting the effects of propolis when used as an intracanal medicament on root dentine microhardness. According to the results of the present study, root dentine microhardness values increased after application of propolis.

In 2020, Gaugouri et al. [33] reported that propolis-enriched chewing gum extract increases the dentine microhardness by facilitating the remineralization of demineralized dentine, enhancing its mineral content, and occluding the dentinal tubules. The present study reported similar findings, that propolis application results in increased dentine microhardness.

According to another study, when propolis was used as a root canal irrigant, it showed the least detrimental effects on the root dentine microhardness [34], corroborating the results of the present study. On the contrary, Elgendy et al. [18] reported that the propolis irrigant has been found to reduce dentine microhardness when compared with sodium hypochlorite and EDTA. A similar study also reported in 2015 that propolis application to root dentine reduces its fracture resistance [35]. Similar to the results of the present study where increased dentine microhardness has been reported after propolis application, an increase in enamel microhardness has also been reported when propolis was applied to the external tooth structure [17,36].

Previously, researchers reported that propolis application to dentine reduces its permeability by partially obliterating the dentinal tubules [37]. Furthermore, in 2005, Sabir et al. [38] found that the bioflavonoids in propolis can form crystals inside the dentinal tubules, thereby facilitating the obliteration of dentinal tubules and reducing the dentinal hypersensitivity. Another study concluded that propolis at pH 8.5 induced occlusion of dentinal tubules [39]. It has also been found that the addition of propolis to CH increased collagen type I expression [40]. These could be possible explanations for the increased microhardness of dentine after propolis application in the present study. An important limitation of the present study is that the mineral content of dentine after the application of endodontic medicaments was not evaluated, which could justify the changes in microhardness values in all the groups, and hence it is recommended to be investigated in future studies.

In the present study, a few limitations have been identified. Firstly, only the short-term effect (for up to 7 days) of both intracanal medicaments was evaluated. Secondly, microhardness testing was performed only on mid-root sections. Thirdly, the principal investigator could not be blinded at the stage of intracanal medicament placement because of the difference in the color of both medicaments. Lastly, merely on the basis of microhardness testing, an assumption of the degree of dentine mineralization cannot be made.

## 5. Conclusions

According to the results of the present study, propolis application on dentine increased its microhardness over time. On the contrary, CH progressively decreased the root dentine microhardness when applied to root canals for up to one week. For this reason, propolis proved to be a better alternative to CH when used as an intracanal medicament without negatively affecting the root dentine mechanical strength. To understand this phenomenon more clearly, more specific tests, such as FTIR and evaluation of the mineral content of dentine after propolis application, must be investigated. Further studies may be performed to evaluate the effects of long-term application of propolis on different levels of the root.

## Figures and Tables

**Figure 1 jfb-14-00144-f001:**
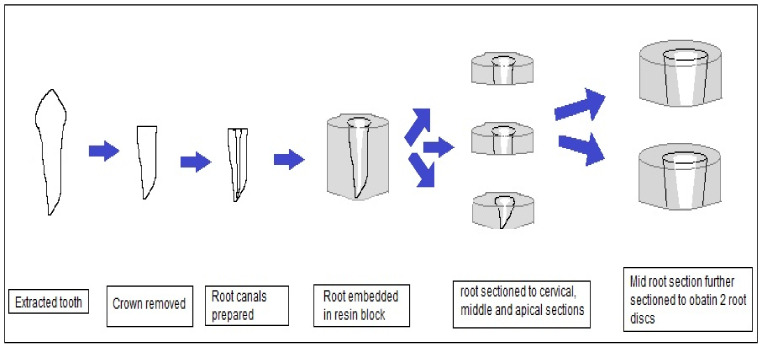
Diagrammatic representation of root dentine discs’ preparation methodology.

**Figure 2 jfb-14-00144-f002:**
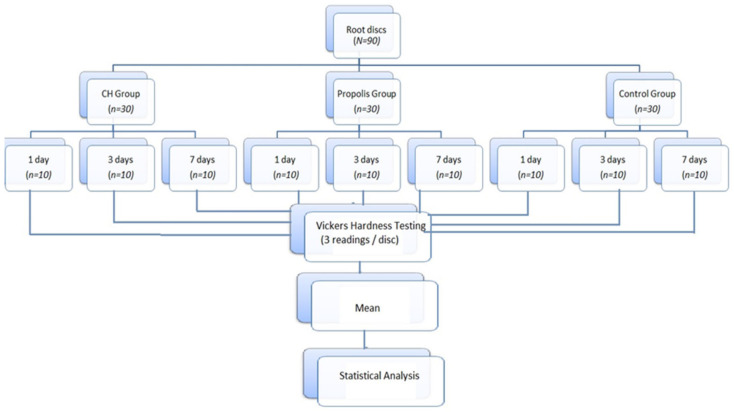
Flow diagram demonstrating a summary of the methodology for the assessment of Vickers hardness of root sections.

**Figure 3 jfb-14-00144-f003:**
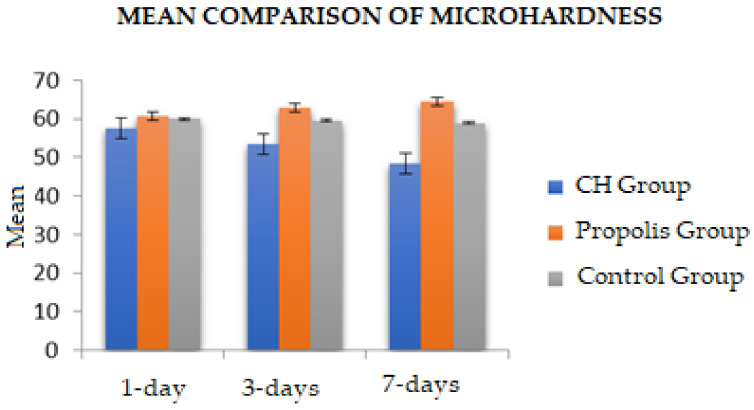
Mean comparison of microhardness values among experimental groups at different time intervals.

**Table 1 jfb-14-00144-t001:** Intragroup comparison of microhardness values (VHN) among experimental groups at different intervals.

Groups	1 Day Mean ± SD	3 Days Mean ± SD	7 Days Mean ± SD	*p*-Value *
Calcium hydroxide (CH)	57.58 ± 1.64	53.53 ± 1.43	48.46 ± 1.60	<0.01 *
Propolis	60.75 ± 1.32	62.97 ± 1.39	64.43 ± 1.69	<0.01 *
Control	59.98 ± 1.27	59.45 ± 1.68	58.87 ± 1.43	0.26

* *p* < 0.05 was considered statistically significant using repeated measures ANOVA.

**Table 2 jfb-14-00144-t002:** Intergroup pairwise comparison of microhardness values (VHN) among experimental groups at 1 day.

Groups	Mean ± SD	*p*-Value
CH vs. Propolis	−3.16 ± 0.66	<0.01 *
CH vs. Control	−2.39 ± 0.65	0.017
Propolis vs. Control	0.76 ± 0.58	0.966

* *p*-value < 0.05 was considered statistically significant (post hoc Tukey test).

**Table 3 jfb-14-00144-t003:** Intergroup pairwise comparison of microhardness values (VHN) among experimental groups at 3 days.

Groups	Mean ± SD	*p*-Value
CH vs. Propolis	−9.42 ± 0.63	<0.01 *
CH vs. Control	−5.90 ± 0.70	<0.01 *
Propolis vs. Control	3.51 ± 0.69	<0.01 *

* *p*-value < 0.05 was considered statistically significant (post hoc Tukey test).

**Table 4 jfb-14-00144-t004:** Intergroup pairwise comparison of microhardness values (VHN) among experimental groups at 7 days.

Groups	Mean ± SD	*p*-Value
CH vs. Propolis	−15.97 ± 0.64	<0.01 *
CH vs. Control	−10.41 ± 0.68	<0.01 *
Propolis vs. Control	5.56 ± 0.70	<0.01 *

* *p*-value < 0.05 was considered statistically significant (post hoc Tukey test).

## Data Availability

Not applicable.

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
