# Peer review of "Effect of Propolis on Root Dentine Microhardness When Used as an Intracanal Medicament: An In Vitro Study"

_jfb, 2023, doi:10.3390/jfb14030144_

Round 1

Reviewer 1 Report

Dear authors,

The introduction needs further improvements by adding more references. A lot of paragraphs with only one reference. Scientifically, this could not be acceptable.

For the materials and methods section the preparation of medicaments needs further explanation, so the reader can better understand.

Figure 1 is out of focus, some good images with acceptable resolution should be done. I didnt like the presentation of this figure.

For the results there is no letter for significance between groups so when the reader read the table he cant be able to see the statistical difference.

Only one limitation was done at the end of the article. However, various limitations should be addressed.

For conclusion section a reformulation must be done for better explanation.

Author Response

Thank you for highlighting the critical mistakes in the manuscript. We have incorporated and improved the manuscript accordingly as follows:

REFREE 1 COMMENTS

1

The introduction needs further improvements by adding more refrences…..

4 refrernces added to introduction (refrence #1,2,5 and 9

2

For materials ans methods section the preparation of medicaments needs further explainations…

Changes have been made in “prep of medicaments” SECTION

3

Figure 1 is out of focus….

Figure 1 has been replaced. A figure of diagramtic representation of root didc preparations have been added for better understanding of the readers

4

For the results, theres no letter of significance….

Results line # 2 stat sig betwe grps mentioned.

Last line of 1st para

3rd para of results, stat sig mentioned again

4th para mentioned

5th para mentioned

5

Only 1 limitation was done….

4 to 5 limitations have been added in the last para of discussion

6

For conclusion section a reformulation ….

Changes in conclusion section have been made.

Reviewer 2 Report

This study evaluated the application of intracanal medicaments (propolis and calcium hydroxide) on the microhardness of root dentin. Of clinical relevance, some aspects of the study need clarification.

1-  In the Introduction authors mention several times, in several instances, terms such as ‘eradication of microbes’ and ‘medicaments eliminate surviving microorganisms. Careful with such statements, they should be replaced/revised as medicaments are not usually able to eliminate microorganisms… 

22- A reference should be added on pg 2 line 85 (where you mention LPS).

 3-      Pg. 3 lines 100 to 104 is a repetition of previous paragraph and should be deleted.

 4-      Pg. 3 Lines 136 to 141: how were these ratios determined. The authors should measure the pH of the solutions used as medicaments and add this information in the manuscript.

 5-      Pg. 4 Data collection procedure: It is unclear if samples were immersed ones or stayed immersed in the intracanal medicaments throughout the testing periods. If the second protocol was followed, what is its clinical relevance? The amount and saturation of solutions would be much higher than what is usually used clinically.

 6-      Tables 2,3, and 4 are not necessary. All the information can be conveyed within Table 1 by using superscript letters within the columns.

 7-      It is hard to make direct assumption on dentin degree of mineralization using one single test such as microhardness. More specific tests such FTIR, and imaging are necessary to properly characterize the treated dentinal tissues.

 8-      Based on previous literature, what is the estimate amount of calcified matrix per square/cubic millimetres in your control group at 24h?

 9-      Pg. 7 Lines 249 to 251: can’t decreased microhardness be also related (at least to some extent) to the softening of the collagen matrix. This needs to be addressed in the discussion. Also, a correlation of possible matrix property alteration with the pH of the medicaments used should be discussed.

 10-   Pg. 8 Lines Lines 291 to 293: the decrease in hardness to CH treatment was only observed at day 7. Clinically, wouldn’t the material be buffered by then? So again, how do your protocol (samples immersed in medicaments?) with clinical treatment?

 11-   Pg. 8 Line 318: it should Gargouri not ‘Wafa’.

 12-   Authors should also discuss what is the effect of propolis on collagen type I? How this interaction can favour maintenance or increase of dentin properties? The pH of the solution can help the discussion.

Author Response

Cover letter

Thank you for highlighting the critical mistakes in the manuscript. We have incorporated and improved the manuscript accordingly as follows:

REFREE 2 COMMENTS

S.NO

COMMENTS

CHANGES

1

In introduction author mentioned several times…..

Suggested changes have been made

2

A ref should be added on pg 2 line 85…

Ref has been added

3

Pg 3 line 100-104 is a repetition….

Suggested changes have been made

4

Pg 3 line 136-141….

Ratios were taken from other studies, ref has been added. Now the pH cannot be determined as the study has finished.

5

Pg 4, data collection….

The samples remained in contact with the medicaments throughout the testing periods as they would be if placed clinically inside the root canals during endodontic therapy.

This methodology was adopted from previous studies used to determine microhardness of extracted teeth when intracanal medicmanets areapplied on teeth.

6

Table 2,3,4 are not necessary…

Table 1 demonstrates intragroup comparison whereas table 2,3 and 4 demonstrate intra group comparisons at different time intervals. Furthermore, table one does not show statistical difference between different groups.  Therefore, for better understanding of our readers, different tables were made.

7

It is hard to make direct assumption….

It is a limitation of this study.Already mentiuoned in the limitations and future directions of this study.

8

Based on previous literature what is the estimated amount of …

Calculation of calcific matrix per square cubis mm is our of scope of this reaserch.

9

Pg 7 line 249 to 251 cant decreased….

Softening of collagen matrix is  already mentioned in 5th para of discussion. Also discussion about pH of both medicaments have been added.

10

Pg 8, line 291 to 293 the decreased hardness…

according to the results dec microhardness is evident on the 3rd day of Ch application aswell. The methodology used in the present study has been adopted from previous studies where same methodology was used for measurement of microhardness testing in extracted teeth. Re #[6]

11

Pg 8 line 318 it is Gargouri not “wafa”

Suggested changes have been made

12

Author should also discuss what is the effect of propolis on collagen type 1….

detailed suggested explainaition has been added to the text in para 9 of discussion.

Round 2

Reviewer 1 Report

Thank you for your revised manuscript. For table 1 a letter of significance between rows and column could be more explicative.

Reviewer 2 Report

Authors have satisfactorily answered the questions.